# Constructing a Glioblastoma Prognostic Model Related to Fatty Acid Metabolism Using Machine Learning and Identifying F13A1 as a Potential Target

**DOI:** 10.3390/biomedicines13020256

**Published:** 2025-01-21

**Authors:** Yushu Liu, Hui Deng, Ping Song, Mengxian Zhang

**Affiliations:** Department of Oncology, Tongji Hospital, Tongji Medical College, Huazhong University of Science and Technology, Wuhan 430030, China; liuyushu9912@163.com (Y.L.); d202382343@hust.edu.cn (H.D.); songping202212@163.com (P.S.)

**Keywords:** glioblastoma, fatty acid metabolism, F13A1, machine learning, prognosis, immune

## Abstract

**Background:** Increased fatty acid metabolism (FAM) is an important marker of tumor metabolism. However, the characterization and function of FAM-related genes in glioblastoma (GBM) have not been fully explored. **Method:** In the TCGA-GBM cohort, FAM-related genes were divided into three clusters (C1, C2, and C3), and the DEGs between the clusters and those in the normal group and GBM cohort were considered key genes. On the basis of 10 kinds of machine learning methods, we used 101 combinations of algorithms to construct prognostic models and obtain the best model. In addition, we also validated the model in the GSE43378, GSE83300, CGGA, and REMBRANDT datasets. We also conducted a multifaceted analysis of F13A1, which plays an important role in the best model. **Results:** C2, with the worst prognosis, may be associated with an immunosuppressive phenotype, which may be related to positive regulation of cell adhesion and lymphocyte-mediated immunity. Using multiple machine learning methods, we identified RSF as the best prognostic model. In the RSF model, F13A1 accounts for the most important contribution. F13A1 can support GBM malignant tumor cells by promoting fatty acid metabolism in GBM macrophages, leading to a poor prognosis for patients. This metabolic reprogramming not only enhances the survival and proliferation of macrophages, but also may promote the growth, invasion, and metastasis of GBM cells by secreting growth factors and cytokines. F13A1 is significantly correlated with immune-related molecules, including IL2RA, which may activate immunity, and IL10, which suggests immune suppression. F13A1 also interferes with immune cell recognition and killing of GBM cells by affecting MHC molecules. **Conclusions:** The prognostic model developed here helps us to further enhance our understanding of FAM in GBM and provides a compelling avenue for the clinical prediction of patient prognosis and treatment. We also identified F13A1 as a possibly novel tumor marker for GBM which can support GBM malignant tumor cells by promoting fatty acid metabolism in GBM macrophages.

## 1. Introduction

Glioma is a common malignant brain tumor, and its incidence is increasing every year [1]. Among gliomas, glioblastoma (GBM) is the most malignant type, accounting for approximately 70–80% of adult glioma cases [2], and is highly likely to metastasize [3]. According to the classification of the World Health Organization (WHO), gliomas are classified into four histopathological grades: Grade I, II, III, and IV. Grades II and III are considered low-grade gliomas (LGGs) and glioblastomas (GBMs, Grade IV), characterized by new angiogenesis, and constitute the most aggressive molecular subtype of glioma. The median overall survival (OS) of GBM patients is 14–16 months and the progression-free survival (PFS) of them is only 6–7 months [4].

In the context of lipid metabolism, particularly fatty acid metabolism (FAM), nutrients undergo transformation into metabolic intermediates, which serve as essential building blocks for membrane synthesis, energy reserve accumulation, and the generation of signaling molecules [5]. Alterations in lipid metabolism are characteristic features and metabolic signatures of cancer cells. Cancer cells significantly adjust their lipid metabolic pathways to meet their rapid growth and division needs. By interfering with the lipid supply of cancer cells, we can have a profound impact on their biological energy processes such as the production and utilization of ATP, the synthesis and maintenance of cell membrane structure, and intracellular signaling mechanisms such as cell proliferation and apoptosis regulated by lipid signaling molecules [6]. Increasing evidence suggests that various alterations take place within tumor tissue throughout distinct stages of FAM [7], influencing the type, abundance, and mode of action of regulatory lipid signaling molecules [8]. These changes jointly shape the metabolic characteristics of tumor cells, providing a new perspective for the occurrence, development, and treatment resistance of tumors. Elevated FAM represents a key aspect of tumor metabolic characteristics [9]. It reflects the urgent need of tumor cells for energy and biosynthetic materials, as well as the metabolic strategies they have formed during the process of adapting to microenvironmental pressures such as hypoxia and nutrient deprivation. However, the characterization and function of FAM-related genes in GBM have not been fully explored. The mechanism of lipid metabolism in GBM, especially the regulatory network of FAM-related genes and its relationship with tumor progression and therapeutic response, remains a frontier topic that needs to be explored urgently.

Therefore, it is important to study the role of FAM in GBM. We used TCGA-GBM data, and GSE43378 and GSE83300 data, to explore the mechanism of action of FAM-related genes in GBM. We divided the patients into three clusters and compared their prognostic and immune characteristics. The differential genes (DEGs) between the normal group and cancer group and DEGs between different clusters were screened as key genes. We also used 10 different machine learning methods, 101 combinations, to build prognostic models and choose the best ones. In addition, we are the first to report that F13A1 plays a very important role in GBM. Considering that different clusters have different immune characteristics, and that the DEGs between different clusters also play a very important role in the immune pathway, we thoroughly analyzed the immunity and function of F13A1 and identified it as a possible prognostic marker of GBM.

In this study, we used FAM-related genes to establish and validate a GBM risk score model via public datasets. On the basis of 10 kinds of machine learning approaches, we used 101 combinations of algorithms to construct prognostic models and obtain the best model. We also conducted a multifaceted analysis of F13A1, which serves as a crucial component in the optimal model, and identified it as a possible tumor marker for GBM.

## 2. Materials and Methods

### 2.1. Data Acquisition

We collected RNA sequencing (RNA-seq) data with full annotation of the clinical information of 168 GBM patients from The Cancer Genome Atlas (TCGA) database (https://portal.gdc.cancer.gov/, accessed on 10 January 2024) and 1148 normal controls were extracted from GTEx (https://www.genome.gov/Funded-Programs-Projects/Genotype-Tissue-Expression-Project, accessed on 14 December 2023). In addition, the validation data were extracted from the Gene Expression Omnibus (GEO) database (GSE43378 (n = 32) and GSE83300 (n = 50); https://www.ncbi.nlm.nih.gov/geo/, accessed on 10 January 2024). Additionally, we downloaded GBM datasets from both the Chinese Glioma Genome Atlas (CGGA) (https://www.cgga.org.cn/, accessed on 27 December 2024) and Repository for Molecular BRAin Neoplasia DaTa (REMBRANDT) (https://gliovis.bioinfo.cnio.es/, accessed on 27 December 2024) databases. We also obtained the GSE84465 raw data of transcriptome profiling from the GEO database.

To obtain FAM-related genes, we used the keyword “fatty acid metabolism” in the Gene Set Enrichment Analysis (http://www.gsea-msigdb.org/gsea/index.jsp, accessed on 5 January 2024) and then retrieved 187 FAM-related genes.

### 2.2. Classification of Molecular Subtypes

The TCGA-GBM and GSE43378 datasets were clustered using the ConsensusClusterPlus package, resulting in the generation of a heat map that visually represents the clustering of the samples. The cumulative distribution function (CDF) was employed to determine the optimal number of clusters and ensure stability in clustering outcomes. Three molecular subtypes, namely, C1, C2, and C3, were subsequently identified. To delve deeper into the prognosis of these different molecular subtypes, Kaplan–Meier (K-M) survival curves were plotted using the survminer package.

### 2.3. Analysis of the Immune Environment

We assessed immune cell infiltration using the R packages MCPcounter, CIBERSORT, quanTIseq, EPIC, and TIMER. Differential analysis among clusters was performed with the R package limma. Stromal and immune scores were derived from the gene expression matrix through the ESTIMATE algorithm. To visualize the analysis results, we utilized heat maps and violin plots.

### 2.4. Filtering of DEGs and Functional Analysis

To identify genes associated with the FAM subtypes, we applied the limma package to compare differences between C1 versus C2 + C3, C2 versus C1 + C3, and C3 versus C1 + C2 in the TCGA-GBM dataset, setting the thresholds at |log2(fold change)| > 1 and FDR < 0.05. GO and KEGG enrichment analyses were subsequently performed on the DEGs. In addition, the DEGs between the GBM and normal groups were intersected with the previous DEGs, and 28 key genes were obtained in total.

### 2.5. Establishment and Validation of the Prognostic Signature

On the basis of 10 machine-learning algorithms ((random survival forest (RSF), Lasso, generalized boosted regression modeling (GBM), CoxBoost, stepwise Cox, partial least squares regression for Cox (plsRcox), supervised principal components (SuperPC), Ridge, elastic network (Enet), and survival support vector machine (survival-SVM)), we used 101 combination algorithms to construct prognostic models on the training set (TCGA-GBM) and testing sets (GSE43378 and GSE83300) based on the leave-one-out cross-validation (LOOCV) framework. Considering the effectiveness, wide applicability, complementarity with other evaluation indicators, robustness of calculation methods, and practicality of clinical application of the C-index evaluation model for predicting prognosis accuracy, we calculated and compared the average C-index of the prognostic model we constructed. RSF was considered to be the best prognostic model. RSF model comprises two parameters: ntree and mtry. Specifically, ntree indicates the total count of trees in the forest, while mtry signifies the number of variables randomly picked for splitting at each node. To find the best combination of these parameters, we implemented a grid-search strategy within the LOOCV framework. This entailed evaluating all possible pairs of (ntree, mtry) and ultimately identifying the pair that produced the highest C-index value as the optimized parameters. Patients were stratified into high-risk and low-risk groups based on the median RiskScore, and a Kaplan–Meier (K-M) curve was plotted to illustrate survival differences. Additionally, a receiver operating characteristic (ROC) curve was generated using the timeROC package. Meanwhile, we also verified the prognostic model we constructed in two independent clinical cohorts, CGGA and REMBRANDT. What is more, we performed a multivariate Cox analysis on the CGGA dataset.

### 2.6. Differential Expression Analysis of F13A1

We compared the RNA expression of F13A1 in normal tissues using the HAP database (https://www.proteinatlas.org/, accessed on 27 July 2024). In addition, we also used the SangerBox database (http://sangerbox.com, accessed on 25 July 2024) to explore the differential expression of F13A1 in pancarcinoma and normal tissues.

### 2.7. Immunity Analysis of F13A1

We used the Timer database (https://cistrome.shinyapps.io/timer/, accessed on 28 July 2024) to investigate the correlation between F13A1 and six types of immune cells and their purity. The CIBERSORT package in R was used to analyze the TCGA-GBM dataset, and the correlation between immune cells and F13A1 is shown by heat maps. According to the median F13A1 expression level, we divided the dataset into high- and low-F13A1 expression groups, and a box plot shows the difference in immune cell infiltration levels between the high- and low-F13A1 expression groups. What is more, we explored the associations of F13A1 with immune regulatory genes in the SangerBox database. Using the TISIDB database (http://cis.hku.hk/TISIDB/, accessed on 28 July 2024), we also explored the associations of F13A1 with immune activators, immunosuppressants, and MHC molecules. The Estimate package was used to investigate the difference in immune infiltration between the high- and low-F13A1 expression groups. Finally, we used the TIDE database (http://tide.dfci.harvard.edu, accessed on 28 July 2024) to evaluate the potential immunotherapeutic effect of immunotherapy in different F13A1 expression subgroups.

### 2.8. Functional Analysis of F13A1

From the STRING database, we downloaded the top 20 most relevant proteins of F13A1. Then, GO and KEGG analyses were subsequently performed. To gain a more comprehensive understanding of the role of F13A1 in GBM, we screened the DEGs between the high- and low-F13A1 expression groups and then performed GSEA analysis on them.

### 2.9. Single-Cell Analysis

We obtained raw transcriptome profiling data, specifically GSE84465, from the GEO database. Initially, we assessed the quality of the single-cell RNA sequencing (scRNA-seq) data using three crucial metrics to establish its reliability: total RNA count, gene diversity, and the proportion of mitochondrial gene expression (termed as nCount_RNA, nFeature_RNA, and percent mt, respectively). Using the anchor method, we integrated the single-cell datasets through the Seurat R package. Subsequently, we refined the data by discarding low-quality scRNA-seq entries to isolate a core set of cells. Cells in which genes were detected in three or fewer cells, as well as those with detections of fewer than 50 genes, indicating poor quality, were excluded. Following this cleanup process, we normalized the single-cell data and performed principal component analysis (PCA) to distinguish various cellular clusters. We named the subpopulations based on the most significantly differentially expressed genes in the cell subpopulations. CytoTRACE and Trajectory analysis were also used for in-depth analysis..

### 2.10. Statistical Analysis

Our study primarily relied on R software(R version 4.4.1) for statistical analyses, defining a *p*-value of less than 0.05 as indicating a statistically significant difference. To evaluate disparities in immune abnormalities among RiskScore groups, we employed the Wilcoxon rank-sum test.

## 3. Results

### 3.1. Identification of Key Genes Associated with FAM

#### 3.1.1. Classification of Three Subtypes Based on FAM Genes

Figure 1 shows the flowchart of the study. We used the ConsensusClusterPlus package to cluster the TCGA-GBM cohort samples (Figure 2A). The CDF curve revealed that three clusters were relatively stable (Figure 2B,C). To further analyze the survival differences across the three clusters, we plotted the K-M curve via the survminer package (*p* = 7.67 × 10^−3^). Compared with C2 and C3, C1 had the best prognosis on the whole (Figure 2D). What is more, we applied the same clustering method to GSE43378 (Figure 2E). The three clusters were still relatively stable (Figure 2F,G) and the K-M curve also revealed significant differences (*p* = 2.588 × 10^−2^). C1 also showed the best prognosis in GSE43378 (Figure 2H). Therefore, we believed that the deep reasons leading to the different prognosis of three clusters are worthy of further exploration.

#### 3.1.2. Analysis of the Immune Microenvironments of the Three Subtypes

Five immunoassay methods, including cibersort, epic, mcp, quantiseq, and timer, were used to explore the immune microenvironment of each cluster (Figure 2I). The heat map revealed significant variations in the levels of immune cell infiltration among the three clusters, potentially impacting the prognosis of GBM patients. The violin diagram shows the results of cibersort analysis (Figure 2J). Naive B cells, plasma cells, activated CD4 memory T cells, follicular helper T cells, T cells regulatory, monocytes,resting mast cells, activated mast cells, and neutrophils showed significant differences in the three clusters. They might be one of the potential factors contributing to the different outcomes of GBM patients. In addition, the Estimate package was used to analyze the immune microenvironment of three clusters (Figure 2K). C1, which exhibited the most favorable prognosis, had the lowest StromalScore, ImmuneScore, and ESTIMATEScore, suggesting an association with an immunoactivated phenotype. Conversely, C2, which had a poor prognosis, demonstrated the highest scores in these categories, indicating an association with an immunosuppressive phenotype.

#### 3.1.3. Functional Analysis of Differential Genes

To investigate the function of DEGs between C1 and C2 + C3, GO and KEGG analyses were performed and the results of the top 10 most significant pathways are shown in Figure 3A,B. The DEGs between C1 and C2 + C3 played an important role in the TNF signaling pathway and PI3K-Akt signaling pathway, which might contribute to the better prognosis of C1. Using the same method, we analyzed the DEGs between C2 and C1 + C3 (Figure 3C,D) and the DEGs between C3 and C1 + C2 (Figure 3E,F). We found that the DEGs between C2 and C1 + C3 were related to positive regulation of cell adhesion and lymphocyte-mediated immunity, which might contribute to a poor prognosis of C2. Furthermore, the DEGs between C3 and C1 + C2 were linked to the Toll-like receptor signaling pathway and the NF-kappa B signaling pathway, potentially contributing to the unfavorable prognosis observed in C3.

#### 3.1.4. Obtaining of Key Genes

To obtain key genes associated with GBM patient prognosis, we identified DEGs between C1 and C2 + C3. By the same method, the DEGs between C2 and C1 + C3, C3 and C1 + C2, as well as between normal group and GBM patients, were obtained. A Venn diagram shows the intersection of these different genes (Figure 3G). These 28 intersection genes were considered to be key genes that played an important role in GBM patient prognosis. The PPI network of key genes in the STRING database is shown in Figure 3H.

### 3.2. Establishment and Validation of the Prognostic Model

#### 3.2.1. Machine Learning

We integrated 10 machine-learning algorithms and 101 combinational algorithms were generated, which were randomly combined together using 10-fold cross-validation approaches. TCGA-GBM was used as the training set, and GSE43378 and GSE83300 were used as the testing sets. The C-index results of machine learning are shown in Figure 4A. In order to compare the models better, we calculated the mean C-index of the datasets (Figure 4B) and RSF was considered to be the best prognostic model. Compared with other models, the RSF model [10] has outstanding performance in terms of interpretability, computational efficiency, and clinical applicability. It is based on decision tree ensemble learning and has a high degree of interpretability, allowing us to evaluate the contribution of each variable through variable importance (VIMP). Our optimizations and parameter settings successfully reduced costs and increased efficiency. In addition, the RSF model has a fast convergence rate and can provide stable prediction results. In clinical terms, it can integrate multiple clinical and genetic indicators to significantly improve the accuracy of predicting the survival of patients with GBM. It can also handle censored survival data, which is crucial for studies involving survival outcomes. The RSF model can better capture the heterogeneity of GBM patients and identify factors closely related to the prognosis of GBM patients. In addition, the RSF model can calculate risk scores, providing clinicians with more intuitive predictive results. Therefore, we choose the RSF model as the best model for subsequent analysis and verification.

#### 3.2.2. RSF Analysis

Figure 4C shows the importance ranking of key genes in RSF, with F13A1 having the highest importance. Additionally, the Sankey chart shows the relationship between three clusters and the RSF risk group, as well as the relationship between the RSF risk groups and patient overall survival (Figure 4D). The findings indicate that patients in the high-risk group had a notably poorer prognosis, and AUC of the RSF model was relatively high (Figure 4E,F). What is more, we also verified RSF in GSE43378, GSE83300, CGGA, and REMBRANDT (Appendix A). Overall, RSF is a relatively reliable prognostic model which might play a guiding role in clinical work. Moreover, we performed a multivariate Cox analysis on the CGGA dataset. The results showed that patients with higher RiskScore had higher risk and worse prognosis (Appendix A).

### 3.3. Analysis of F13A1

#### 3.3.1. Differential Expression Analysis of F13A1

Considering that F13A1 had the highest importance in the RSF model, we hypothesized that F13A1 could play a pivotal role in the initiation and progression of GBM, which was very worthy of our in-depth exploration. We used the GTEx dataset from the HAP database to investigate the distribution of RNA expression of F13A1 in normal tissues (Figure 5A). The result showed that RNA expression of F13A1 in adipose tissue was the highest. In addition, the difference in expression of F13A1 in pancarcinoma and normal tissues was also statistically analyzed (Figure 5B). We found that the expression of F13A1 in most cancers was statistically different from that in normal tissues. The expression of F13A1 was up-regulated in some cancers, including GBM, LGG, KIRC, and so on. The expression of F13A1 was down-regulated in other cancers including UCEC, BRCA, CESC, and so on. This suggested that F13A1 might play a different role in different cancers. In addition, we obtained Western Blot (WB) experimental images of F13A1 in various cell lines from the HAP database, and the results showed that F13A1 was highly expressed in the U251 MG cell line (Appendix A).

#### 3.3.2. Immunity Analysis of F13A1

To delve deeper into the potential immune-related functions of F13A1 in GBM, we used the Timer database to analyze its correlation with six types of immune cells and purity (Figure 5C). F13A1 was positively correlated with CD4+T cells (partial.cor = 0.104, *p* = 3.36 × 10^−2^), macrophages (partial.cor = 0.133, *p* = 6.28 × 10^−3^), neutrophils (partial.cor = 0.154, *p* = 1.60 × 10^−3^), and dendritic cells (partial.cor = 0.485, *p* = 4.37 × 10^−26^). Additionally, we used the cibersort package in R to analyze the TCGA-GBM dataset, and the correlation between immune cells and F13A1 is shown by a heat map (Figure 5D). F13A1 demonstrated the highest positive correlation with activated CD4 memory T cells (partial correlation coefficient = 0.37) and the strongest negative correlation with follicular helper T cells (partial correlation coefficient = −0.37). Based on the median expression level of F13A1, we stratified the dataset into high- and low-F13A1 expression groups. The infiltration degrees of activated CD4 memory T cells, follicular helper T cells, activated NK cells, monocytes, macrophages M0, resting mast cells, and neutrophils were statistically different in the high- and low-F13A1 expression groups (Figure 5E). Using the SangerBox database, we also analyzed the correlation between F13A1 and immune regulatory genes (Figure 5F). We found that F13A1 was positively correlated with most immunomodulatory genes with statistical significance in TCGA-GBM, suggesting that F13A1 plays an important immune role in GBM.

Therefore, we conducted a more in-depth analysis of the relationship between F13A1 and immune regulatory genes. The correlation between the expression of F13A1 and immunostimulator (Figure 6A), immunoinhibitor (Figure 6B), and MHC molecules (Figure 6C) in pancarcinoma is demonstrated by a heat map. In GBM, the immunostimulator with the highest correlation with F13A1 was IL2RA (Figure 6D). The immunoinhibitor with the highest correlation with F13A1 was IL10 (Figure 6E), and the MHC molecule with the highest correlation with F13A1 was HLA-DRA (Figure 6F).

For investigating the immune function of F13A1 as a whole, we used estimate for analysis. The results showed that F13A1 was positively correlated with ImmuneScore (r = 0.62, *p* = 9.3 × 10^−18^) (Figure 6G), StromalScore (r = 0.74, *p* = 7.8 × 10^−28^) (Figure 6H), and ESTIMATEScore (r = 0.70, *p* = 1.1 × 10^−23^) (Figure 6I). Additionally, the scores for the group with elevated F13A1 expression were greater than those for the group with lower F13A1 expression (Figure 6J), which suggested that high expression of F14A1 might be associated with immunosuppressive phenotype. Subsequently, we utilized the TIDE database to assess the potential efficacy of immunotherapy in various F13A1 expression subgroups (Figure 6K). Higher TIDE prediction scores signify a greater tendency for immune evasion, implying that patients may be less responsive to ICI treatment. The high-F13A1 expression group had a high TIDE score and dysfunction score, while the low-F13A1 expression group had a high exclusion score. MSI was inconvenient to compare due to the small number of data. Overall, these results suggested that high expression of F13A1 would lead to reduced benefits of immunotherapy in patients with GBM.

#### 3.3.3. Functional Analysis of F13A1

First, we downloaded the top 20 most relevant proteins of F13A1 from the STRING database, as shown in Figure 7A. Then, we conducted GO analysis (Figure 7B) and KEGG analysis (Figure 7C) on them and showed the top five pathways. To gain a more thorough insight into the role of F13A1 in GBM, we identified the DEGs between the high- and low-F13A1 expression groups (Figure 7D) and subsequently conducted GSEA analysis on these genes (Figure 7E). We found that the DEGs were involved in DNA replication, ribosomes, and nucleocytoplasmic transport, and so on. Following that, we separately performed GSEA analysis on the up-regulated and down-regulated genes (Figure 7F). Notably, the down-regulated genes were significantly involved in cell cycle regulation and DNA replication, hinting at a crucial role of F13A1 in the initiation and progression of GBM.

#### 3.3.4. F13A1 Is Specifically Highly Expressed in Macrophages

In the single-cell sequencing dataset GSE84465, we identified seven different types of cells, which include immune cell, oligodendrocytes, vascular, neurons, astrocytes, OPCs, and neoplastic (Figure 8A).

Interestingly, we observed that F13A1 is highly expressed in immune cells (Figure 8B,C). We then extracted the immune cells for more detailed annotation (Figure 8D). Thus, we found that F13A1 was mainly expressed in macrophages (Figure 8E). In order to further understand the role of F13A1 in macrophages, we classified macrophages into subgroups (Figure 8F). F13A1 is mainly highly expressed in cluster 1 and cluster 4 (Figure 8G). According to the most differentially expressed genes in the macrophage subpopulations, we named the subpopulations as INHBA+ Macrophages (cluster 0), E4F1+ Macrophages (cluster 1), ABCG2+ Macrophages (cluster 2), ERCC+ Macrophages (cluster 3), and CXCL1+ Macrophages (cluster 4) (Figure 8H,I). In addition, we also investigated the differentiation status of different macrophage subpopulations. Using CytoTRACE to calculate the differentiation status of each macrophage subpopulation, we found that the E4F1+ macrophages had the lowest differentiation score (Figure 8J). F13A1 shows the centralization of differentiation in E4F1+ macrophages (Figure 8K&8L). Therefore, we speculate that F13A1 may be able to influence the differentiation of macrophages, and then affect the process of GBM.

## 4. Discussion

GBM, a highly lethal and therapy-resistant primary brain tumor, has prompted extensive research into its metabolic mechanisms to discover innovative treatment strategies. Among them, FAM plays a key role in the pathogenesis of GBM [11]. There are reports indicating that lipid metabolism disorder in GBM is associated with prognosis deterioration [12,13]. Thus, investigating the impact of FAM on GBM prognosis holds significant clinical importance. In recent years, the combination of metabolic targets and immunotherapy has shown potential to improve clinical outcomes. Among various metabolic processes, FAM is vital for cancer cell survival and proliferation and plays a crucial role in the differentiation and migration of tumor-associated immune cells. We classified FAM-related genes and found that the subtype with the worst prognosis, C2, may be associated with an immunosuppressive phenotype and positive regulation of cell adhesion and lymphocyte-mediated immunity. Notably, significant differences were observed in CD4+ memory T cells across the three subtypes. Daniel Dubinski and colleagues demonstrated that CD4+ T memory cell dysfunction is correlated with the accumulation of granulocytic myeloid-derived suppressor cells in GBM patients [14].

Over the past ten years, machine learning has witnessed a surge in medical applications, particularly within the field of oncology [15]. We utilized ten machine learning techniques to develop prognostic models based on key genes and identified RSF as the best model. There are four genes (F13A1, SERPINE1, MXRA5, and RARRES1) closely tied to the prognosis of GBM. Previous research shows SERPINE1 as a key gene in glioma proliferation and metastasis. Wang Z et al. found that CAV-1 up-regulates SERPINE1, activating the PI3K/Akt pathway, which promotes glioma growth and spread via epithelial–mesenchymal transition (EMT) and angiogenesis [16]. Moreover, activation of the PAI-1/LRP1 axis, involved in mast cell recruitment in gliomas, enhances STAT3 phosphorylation and exocytosis [17]. SERPINE1 also has oncogenic potential in hepatocellular [18], gastric [19], and vulvar cancers [20]. MXRA5, a glycoprotein in the MXRA protein family, affects cell adhesion and extracellular matrix (ECM) remodeling [21]. Prior studies have demonstrated that high MXRA5 expression in GBM, especially the more invasive mesenchymal subtype with poorer prognosis [22]. Bioinformatic analysis by Rahane CS et al. supports MXRA5 as a treatment target for GBM [23]. Han X et al. also suggest MXRA5 as a potential immunotherapy target for GBM patients [24]. In other cancers, MXRA5 has also been shown to be a cancer-promoting gene [25,26,27]. RARRES1 levels are linked to CpG island methylator phenotype (CIMP) status, IDH1 mutation, and MGMT methylation, suggesting a connection to malignant GBM [28]. RARRES1 may interact with AQP4, affecting GBM prognosis [29]. Although the mechanism of RARRES1 in GBM is still unclear, its mechanism in many other cancers has been studied in depth [30,31]. For example, in renal clear cell carcinoma, RARRES1 plays an anti-tumor role by promoting ICAM1 expression and inducing the activation of M1 macrophages [32]. Overall, SERPINE1, MXRA5, and RARRES1 have all been shown to be closely related to the immune, prognostic, or targeted treatment of GBM.

The fibrin-stabilizing factor F13A1 forms a heterotetrameric protein complex [33]. High F13A1 expression in lung cancer patients may contribute to tumor metastasis [34] by hindering the clearance of micrometastatic tumor cells mediated by NK cells [35]. Elevated expression of F13A1 in inflammatory monocytes of lung squamous carcinoma patients enhances fibrin cross-linking, promoting lung cancer cell metastasis [36]. Additionally, increased plasma F13A1 activity is also observed in patients with non-small cell lung cancer [37]. Furthermore, F13A1 is linked to poor outcomes in acute promyelocytic leukemia [38]. However, studies on F13A1 in GBM are limited. Lehrer S et al. found a lower tumor copy number of F13A1 is associated with poorer survival in glioblastoma [39], while Zhang Y et al. suggested F13A1 as a candidate for studying N6-methyladenosine-mediated GBM development [40]. Among the four genes in our constructed model, we chose to focus on F13A1. We are the first to conduct in-depth research on the mechanism of F13A1 in GBM.

Our study novelly analyzed F13A1’s immune mechanisms and functions in GBM, finding it up-regulated with increasing malignancy and associated with poorer prognosis. As fatty acids are key bioenergetic substrates in glioma cells [41], we built a GBM prognostic model using key fatty acid metabolism genes, highlighting F13A1’s significance. Prior studies link F13A1 to body mass index (BMI) [42] and weight gain [43], and it correlates with gene transcripts related to adipocyte size, linked to fatty acid metabolism. Thus, we hypothesize that F13A1 promotes GBM by affecting fatty acid metabolism, though mechanisms are unclear.

There were statistically significant differences in the infiltration levels of various immune cell types between the high- and low-expression groups of F13A1, suggesting that there are significant differences in the immune microenvironment between the high- and low-expression groups of F13A1. At the same time, patients with high expression of F13A1 in GBM are associated with worse prognosis. We speculate that the immune cells in the high- and low-expression groups of F13A1 play a pivotal role in the prognosis of GBM patients. Based on single-cell sequencing, we found that F13A1 is mainly expressed in malignant cells and macrophages, especially macrophages. Macrophages play an important role in the tumor microenvironment of GBM. They not only participate in physiological processes such as inflammation regulation and tissue repair, but also support the growth and invasion of tumor cells through metabolic reprogramming, which further supports the importance of macrophages in the pathogenesis of GBM. Meanwhile, through GSEA analysis, we found that F13A1 may affect GBM patients by affecting cell cycle and DNA replication. Research has shown that the interaction between brain fatty acid binding proteins and the polyunsaturated fatty acid environment is a potential determinant of poor prognosis in malignant gliomas [44]. The study by Suganami T et al. [45] found that co-culture of differentiated adipocytes and macrophages induces inflammatory changes. Macrophage-derived TNF-α increases the release of free fatty acids (FFAs) in adipocytes, which in turn induces inflammatory changes in macrophages at least in part through the MAP kinase pathway. Therefore, we propose a conjecture that F13A1 can support GBM malignant tumor cells by promoting fatty acid metabolism in GBM macrophages, leading to poor prognosis for patients. However, the specific mechanism of F13A1 promoting macrophage fatty acid metabolism is still unclear, and further research experiments are needed.

Our research reveals a crucial role of F13A1 in GBM progression: it supports GBM cells by enhancing fatty acid metabolism in macrophages. Specifically, F13A1 may increase the metabolic activity of macrophages by regulating metabolic pathways within them, such as fatty acid β-oxidation, thereby providing necessary metabolic support, such as energy and metabolic intermediates, to GBM cells. This metabolic reprogramming not only enhances the survival and proliferation capabilities of macrophages but also potentially promotes the growth, invasion, and metastasis of GBM cells through the secretion of growth factors, cytokines, and other factors. F13A1 correlates significantly with immune-related molecules, including IL2RA, which may activate immunity, and IL10, hinting at immune suppression. HLA-DRA expression changes reflect immune status alterations in GBM patients, emphasizing F13A1’s importance in the GBM immune microenvironment. High F13A1 expression reduces immunotherapy benefits by altering the immune balance, inhibiting immune cell activation and antitumor effects. F13A1 also interferes with immune cell recognition and killing of GBM cells by affecting MHC molecules. These findings unveil a new mechanism of F13A1 in GBM and offer fresh insights into immunotherapy and targeted therapy.

Future research can further explore the interaction mechanism between F13A1 and immune cells, and how to optimize the strategy of immunotherapy by targeting F13A1. By inhibiting the function of F13A1, we can block the metabolic support of macrophages for tumor cells, thereby slowing down the growth and progression of GBM. This treatment strategy may have synergistic effects with other treatment modalities such as surgery, radiotherapy, and chemotherapy, improving the therapeutic effect of GBM. Of course, to achieve this goal, we still need to conduct a large amount of experimental verification and clinical research. First, we need to clarify the specific mechanism of F13A1 in GBM, including how it regulates macrophage metabolism and how this regulation affects the growth and invasion of tumor cells. Secondly, we need to develop effective F13A1 inhibitors to block its mediated immunosuppressive effects, or enhance the efficacy of immunotherapy by combining immunotherapy drugs with F13A1 inhibitors, and evaluate their safety and effectiveness in the treatment of GBM. Finally, we also need to conduct large-scale clinical trials to verify the potential and feasibility of targeting F13A1 for the treatment of GBM.

Our research does have its limitations. Although we validated the prognostic model across multiple datasets, these datasets may not fully represent the characteristics of all GBM patients. As a result, the universality and accuracy of the model may be limited to some extent. Various machine learning methods are used to build and validate models, but the process of model construction and validation may be affected by factors such as algorithm selection and parameter settings. Moreover, although the analysis of F13A1 is comprehensive, there may still be other aspects or potential interaction mechanisms that have not been considered. Even though we have found the correlation between F13A1 and immune-related molecules, we still lack in-depth exploration of how F13A1 specifically affects the function and mechanism of the immune system. The article is mainly based on bioinformatic analysis, and lacks experimental verification to support the function and mechanism of F13A1 in GBM. Prognostic models provide clinical prediction pathways for the prognosis and treatment of GBM patients, but the practical application of the models may be affected by various factors, such as individual differences in patients and diversity of treatment regimens. The applicability and accuracy of the model need to be carefully considered in clinical applications. Consequently, our article provides a new perspective and approach for the prognosis and treatment of GBM, but there are still some limitations. Future research can further explore and improve these aspects to improve the accuracy and clinical application value of the model.

## 5. Conclusions

In summary, we analyzed the expression of FAM-related genes in GBM and classified them into three clusters (C1, C2, and C3). Among them, C2 has the worst prognosis, which may be related to the positive regulation of cell adhesion and lymphocyte-mediated immunity. Using multiple machine learning methods, we identified RSF as the best prognostic model. In the RSF model, F13A1 accounts for the most important contribution. The prognostic model developed here helps us to further enhance our understanding of FAM in GBM and provides a compelling avenue for the clinical prediction of patient prognosis and treatment. We also identified F13A1 as a possibly novel tumor marker for GBM which can support GBM malignant tumor cells by promoting fatty acid metabolism in GBM macrophages.

## Figures and Tables

**Figure 1 biomedicines-13-00256-f001:**
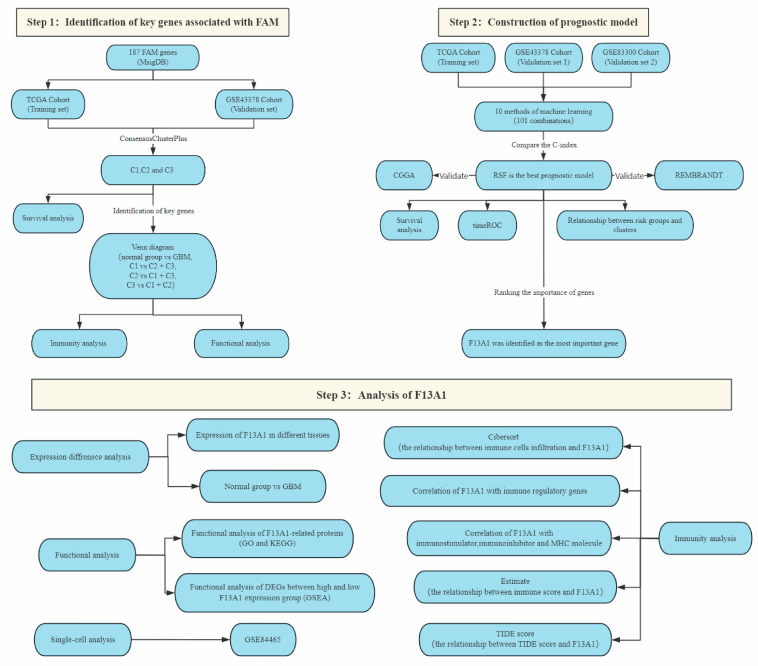
The flowchart of the study.

**Figure 2 biomedicines-13-00256-f002:**
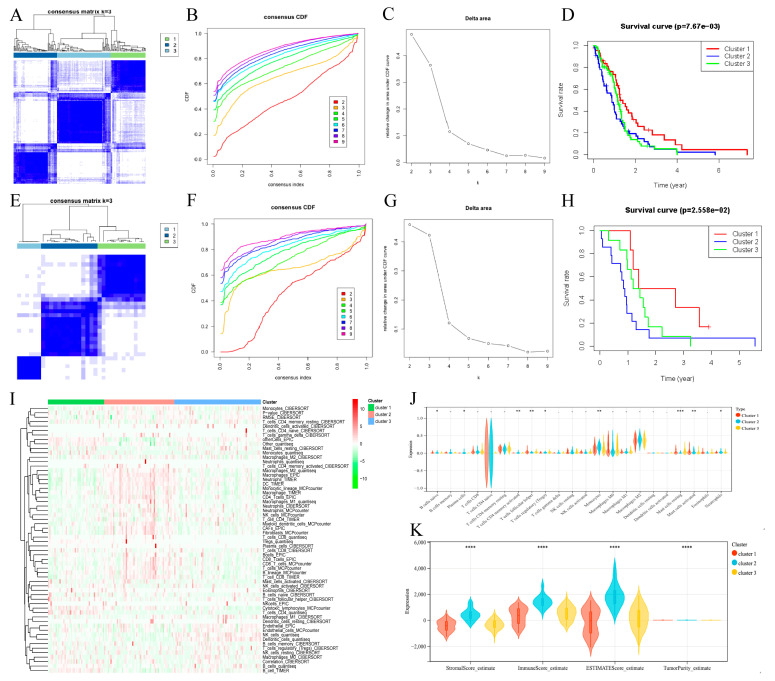
Construction of molecular subtypes based on FAM-related genes. (**A**) Clustering heat map of samples in the TCGA-GBM cohort when consensus k = 3. (**B**) CDF curve of TCGA-GBM cohort samples. (**C**) Delta area of TCGA-GBM cohort samples. (**D**) Kaplan–Meier OS analysis of three clusters in TCGA-GBM cohort samples (*p* = 7.67 × 10^−3^). (**E**) Clustering heat map of samples in the GSE43378 cohort when consensus k = 3. (**F**) CDF curve of GSE43378 cohort samples. (**G**) Delta area of GSE43378 cohort samples. (**H**) Kaplan–Meier OS analysis of three clusters in GSE43378 cohort samples (*p* = 2.558 × 10^−2^). (**I**) The heat map shows the immune microenvironment of different clusters. (**J**) The violin diagram shows the results of cibersort analysis. (**K**) The violin plot shows the immune scores of different clusters. * indicates *p* < 0.05, ** indicates *p* < 0.01, *** *p* < 0.001, **** *p* < 0.0001.

**Figure 3 biomedicines-13-00256-f003:**
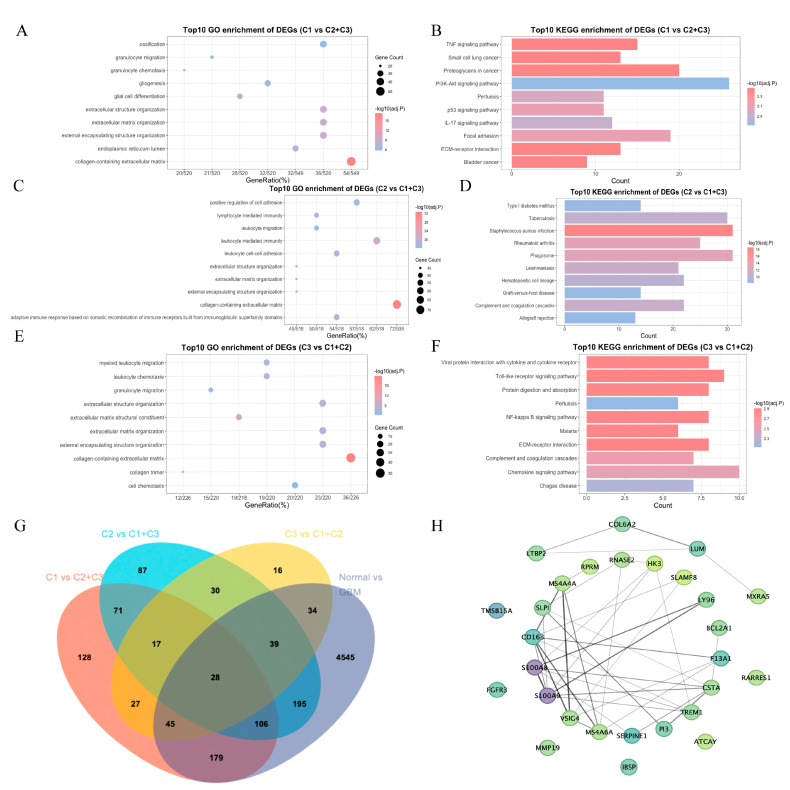
Functional analysis of differential genes. (**A**) Bubble diagram of the top ten functional enrichment analyses of differential genes (C1 vs. C2 + C3) from GO enrichment analysis. (**B**) Histogram of the top ten functional enrichment analyses of differential genes (C1 vs. C2 + C3) from KEGG enrichment analysis. (**C**) Bubble diagram of the top ten functional enrichment analyses of differential genes (C2 vs. C1 + C3) from GO enrichment analysis. (**D**) Histogram of the top ten functional enrichment analyses of differential genes (C2 vs. C1 + C3) from KEGG enrichment analysis. (**E**) Bubble diagram of the top ten functional enrichment analyses of differential genes (C3 vs. C1 + C2) from GO enrichment analysis. (**F**) Histogram of the top ten functional enrichment analyses of differential genes (C3 vs. C1 + C2) from KEGG enrichment analysis. (**G**) Venn diagram to screen for key genes. (**H**) The PPI network of key genes in the STRING database.

**Figure 4 biomedicines-13-00256-f004:**
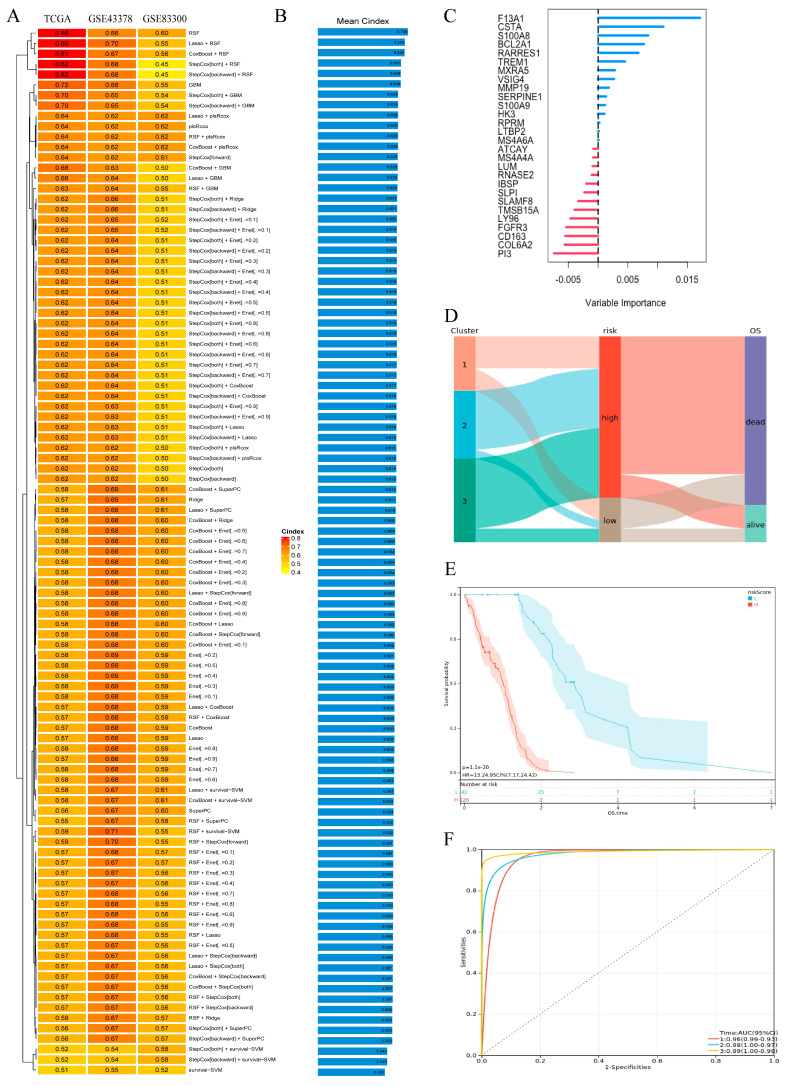
Establishment and validation of the prognostic model. (**A**) The C-index of the prognostic models. (**B**) The mean C-index of each model across all datasets. (**C**) Variable importance in RSF. (**D**) Sankey diagram showing the relationship between cluster, risk, and OS. (**E**) Kaplan–Meier OS analysis of three clusters in TCGA-GBM cohort samples (*p* = 1.1 × 10^−20^). (**F**) Time-dependent ROC analysis for predicting OS at 1, 2, and 3 years in TCGA-GBM cohort samples.

**Figure 5 biomedicines-13-00256-f005:**
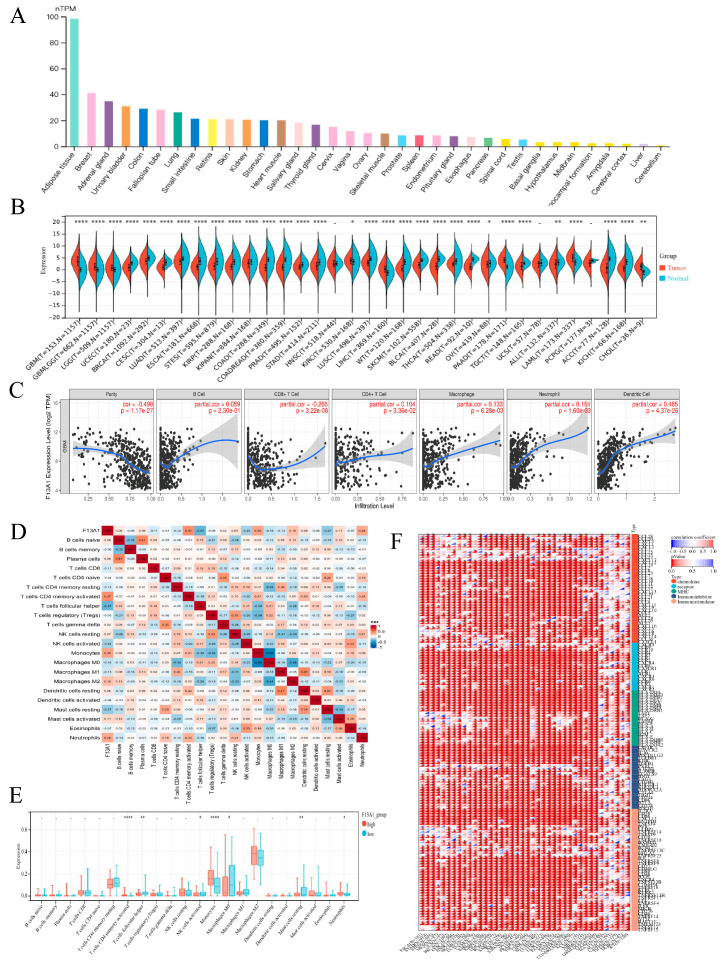
Differential expression analysis of F13A1. (**A**) The RNA expression of F13A1 in normal tissues. (**B**) The RNA expression of F13A1 in different cancers compared with normal group. (**C**) The correlation between F13A1 and six types of immune cells and purity. (**D**) The heat map shows the correlation between F13A1 and immune cells. (**E**) Comparing the expression of immune cells in the high- and low-F13A1 group. (**F**) The heat map shows the correlation between F13A1 and immune regulatory genes.* indicates *p* < 0.05, ** indicates *p* < 0.01, **** *p* < 0.0001.

**Figure 6 biomedicines-13-00256-f006:**
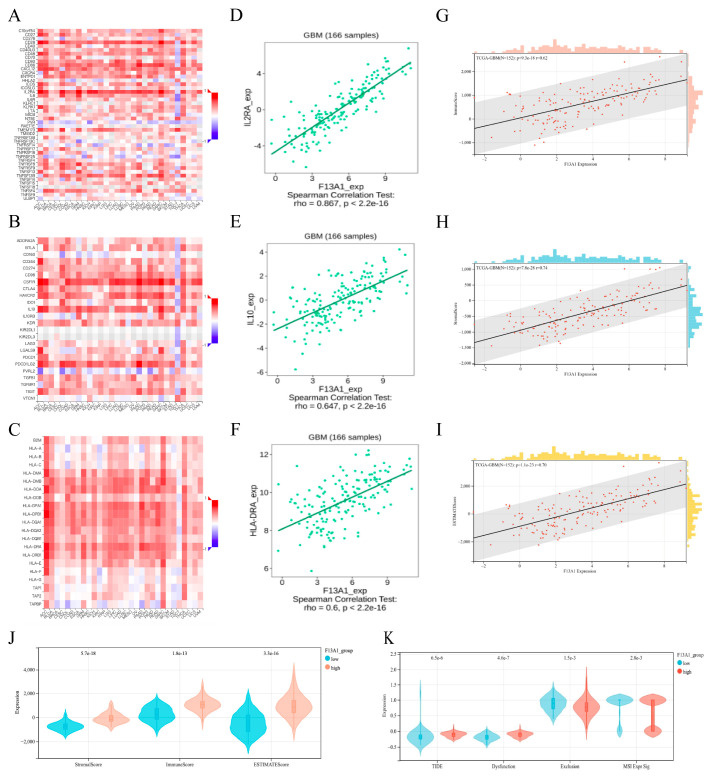
Immunity analysis of F13A1. (**A**) The heat map shows the correlation between F13A1 and immunostimulator in different cancers. (**B**) The heat map shows the correlation between F13A1 and immunoinhibitor in different cancers. (**C**) The heat map shows the correlation between F13A1 and MHC molecules in different cancers. (**D**) The scatter plot shows the relationship between F13A1 and IL2RA in GBM. (**E**) The scatter plot shows the relationship between F13A1 and IL10 in GBM. (**F**) The scatter plot shows the relationship between F13A1 and HLA-DRA in GBM. (**G**) The scatter plot shows the relationship between F13A1 and ImmuneScore in GBM. (**H**) The scatter plot shows the relationship between F13A1 and StromalScore in GBM. (**I**) The scatter plot shows the relationship between F13A1 and ESTIMATEScore in GBM. (**J**) Comparing the immune score in the high- and low-F13A1 group. (**K**) Comparing the TIDE prediction scores in the high- and low-F13A1 expression group.

**Figure 7 biomedicines-13-00256-f007:**
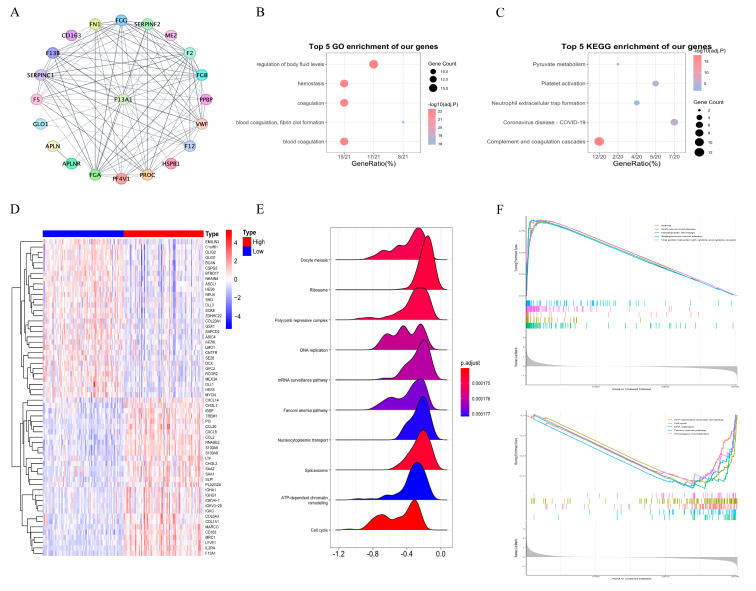
Functional analysis of F13A1. (**A**) Top twenty most relevant proteins of F13A1. (**B**) Bubble diagram of the top five functional enrichment analyses of F13A1-related proteins from GO enrichment analysis. (**C**) Bubble diagram of the top five functional enrichment analyses of F13A1-related proteins from KEGG enrichment analysis. (**D**) Heatmap of differential genes between the high- and low-F13A1 expression group. (**E**) The mountain map shows the top ten most significant pathways in the GSEA analysis of differential genes. (**F**) The top five most significant pathways in the GSEA analysis results of up-regulated genes and down-regulated genes.

**Figure 8 biomedicines-13-00256-f008:**
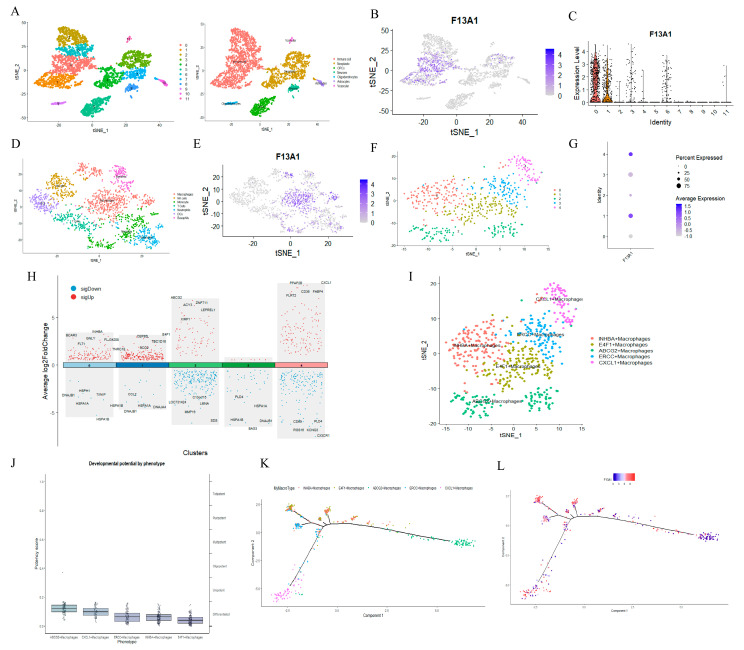
Single-cell analysis of F13A1. (**A**) Cell clustering and identification by tSNE in GSE84465. (**B**) Localization of F13A1 in cell population. (**C**) Expression level of F13A1 in each cluster. (**D**) Cell clustering and identification of immune cells. (**E**) Localization of F13A1 in immune cells. (**F**) Cell clustering and identification of macrophages. (**G**) Expression level of F13A1 in macrophage subpopulation. (**H**) The top 5 most significant differential genes in each macrophage subpopulation. (**I**) Distribution of macrophage subpopulations. (**J**) CytoTRACE score of each macrophage subpopulation. (**K**) Trajectory analysis of macrophages. (**L**) Trajectory analysis of F13A1.

## Data Availability

The original contributions presented in the study are included in the article and Appendix A, further inquiries can be directed to the corresponding author.

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
