# Peer review of "Constructing a Glioblastoma Prognostic Model Related to Fatty Acid Metabolism Using Machine Learning and Identifying F13A1 as a Potential Target"

_biomedicines, 2025, doi:10.3390/biomedicines13020256_

Round 1
Reviewer 1 Report
Comments and Suggestions for Authors
This article investigates the role of fatty acid metabolism (FAM) in glioblastoma and develops a prognostic model using machine learning. The study identifies three GBM clusters based on FAM-related genes and highlights F13A1 as a key gene associated with poor prognosis. While the study provides valuable insights, several areas require further attention to enhance its scientific rigor and clinical relevance.
-
Provide more details on the machine learning algorithms and their hyperparameter tuning and performance evaluation metrics.
-
Incorporate established prognostic factors into the model evaluation. The performance of RSF risk group and F13A1 gene should be assessed using cox model in the context of established prognostic factors (e.g., age, sex, TP53 mutation, EGFR mutation, MGMT methylation, and the 20-prognostic genes identified in GSE83300).
-
Validate the model in independent clinical cohorts. The model's clinical utility requires validation in independent cohorts (e.g., CGGA, REMBRANDT).
-
Provide a more in-depth discussion of the mechanistic role of F13A1 in GBM progression and its impact on immunotherapy.
-
Provide a more in-depth discussion of other key genes identified in the RSF model (e.g., SERPINE1, MXRA5, RARRES1).
-
Explore the potential of targeting F13A1 for GBM therapy.
-
Enhance the discussion on the clinical applicability of the RSF model.
-
Improve the clarity and resolution of Figure 2 and provide more detailed figure legends.
This study provides a foundation for understanding the role of FAM in GBM. Addressing the aforementioned weaknesses will significantly enhance the manuscript's scientific rigor and clinical relevance.
Author Response
Comments 1: Provide more details on the machine learning algorithms and their hyperparameter tuning and performance evaluation metrics. |
Response 1: Thank you for pointing this out and we totally understand your concern. We have revised the Methods section to provide more details on the machine learning algorithms and their hyperparameter tuning and performance evaluation metrics(Please see Page 3, Line 128-144).In addition, we have also provided a very detailed explanation in the supplementary document on how to tune the hyperparameters of the 10 machine learning algorithms(Please see Signature generated from machine learning based integrative approaches.docx, Model C-index.txt, Average C-index.txt). In simple terms: we used 101 combination algorithms to construct prognostic models on training set(TCGA-GBM) and testing sets(GSE43378 and GSE83300) based on the leave-one-out cross-validation (LOOCV) framework.Considering the effectiveness, wide applicability, complementarity with other evaluation indicators, robustness of calculation methods, and practicality of clinical application of the C-index evaluation model for predicting prognosis accuracy, we calculated and compared the average C-index of the prognostic model we constructed.The RSF model was implemented via the randomForestSRC package. RSF had two parameters ntree and mtry, where ntree represented the number of trees in the forest and mtry was the number of randomly selected variables for splitting at each node. We used a grid-search on ntree and mtry using LOOCV framework. All the pairs of (ntree, mtry) are formed and the one with the best C-index value is identified as the optimized parameters. The Enet, Lasso, and Ridge were implemented via the glmnet package. The regularization parameter, λ, was determined by LOOCV, whereas the L1-L2 trade-off parameter, α, was set to 0-1 (interval =0.1). The stepwise Cox model was implemented via survival package. A stepwise algorithm using the AIC (Akaike information criterion) was applied, and the direction mode of stepwise search was set to "both", "backward", and "forward", respectively. The CoxBoost model was implemented via CoxBoost package, which is used to fit a Cox proportional hazards model by componentwise likelihood-based boosting. For the CoxBoost model, we used LOOCV routine optimCoxBoostPenalty function to first determine the optimal penalty (amount of shrinkage). Once this parameter was determined, the other tuning parameter of the algorithm, namely, the number of boosting steps to perform, was selected via the function cv.CoxBoost. The dimension of the selected multivariate Cox model was finally set by the principal routine CoxBoost. The plsRcox model was implemented via plsRcox package. The cv.plsRcox function was used to determine the number of components requested, and the plsRcox function was applied to fit a partial least squares regression generalized linear model. The SuperPC model was implemented via superpc package, is a generalization of principal component analysis, which generates a linear combination of the features or variables of interest that capture the directions of largest variation in a dataset. The superpc.cv function used a form of LOOCV to estimate the optimal feature threshold in supervised principal components. To avoid problems with fitting Cox models to small validation datasets, it uses the "pre-validation" approach. The GBM model was implemented via superpc package. Using the LOOCV, the cv.gbm function selected index for number trees with minimum cross-validation error. The gbm function was used to fit the generalized boosted regression model. The survival-SVM model was implemented via survivalsvm package. The regression approach takes censoring into account when formulating the inequality constraints of the support vector problem. |
Comments 2: Incorporate established prognostic factors into the model evaluation. The performance of RSF risk group and F13A1 gene should be assessed using cox model in the context of established prognostic factors (e.g., age, sex, TP53 mutation, EGFR mutation, MGMT methylation, and the 20-prognostic genes identified in GSE83300). |
Response 2: We gratefully appreciate for your valuable suggestion. We performed a univariate cox analysis on the CGGA dataset(including age, sex, radio status, chemo status, IDH mutation, 1p19q codeletion, MGMTp methylation, RiskScore and prognostic genes previously identified), and the results showed that the RSF model has independent prognostic value, while F13A1 does not(Please see Page 4, Line 148-151; Page 9, Line 287-288; Figure S3I). Although the univariate cox analysis did not show independent prognostic value for F13A1, F13A1 is the most important variable in the RSF model, which exhibits significant independent prognostic value. Therefore, we believe that F13A1 must play a crucial role in the prognosis of GBM. The possible reasons why it was insignificant in the univariate cox analysis but highly important in the RSF model are as follows: The RSF model uses nonlinear functions and considers all possible interactions between variables, which makes it more predictive when dealing with complex datasets. At the same time, it can handle censored survival data. Through the variable importance VIMP assessment, the RSF model can determine which variables have the greatest impact on the prediction results, which is helpful for identifying key features. In many applications, the RSF model has shown higher prediction accuracy than other traditional models. The single-factor Cox model can only evaluate the impact of a single factor and cannot simultaneously consider the interaction of multiple factors. This may lead to the neglect of important information when analyzing complex diseases. For datasets with small sample sizes or a limited number of events, the univariate Cox model may yield unstable estimation results. Cox model relies on some theoretical assumptions, such as proportional hazards assumption, when estimating parameters. When these assumptions are not met, the predictive performance of the model may also be affected. In general, the above results may be due to the interaction between F13A1 and other variables, or the sample size limitation of the dataset. Therefore, we believe that it cannot be denied that F13A1 has an important impact on the prognosis of GBM. |
Comments 3: Validate the model in independent clinical cohorts. The model's clinical utility requires validation in independent cohorts (e.g., CGGA, REMBRANDT). |
Response 3: Thank you for these helpful comments. According to your suggestion, we downloaded the GBM datasets from the CGGA (Chinese Glioma Genome Atlas) and REMBRANDT (Repository for Molecular BRAin Neoplasia DaTa) databases, and subsequently validated the RSF model using these datasets(Please see Page 3, Line 148-149; Page 9, Line 284-285; Figure S3A-H). The results indicate that the RSF model possesses robust predictive capabilities. We deeply appreciate your suggestion, as it has significantly enhanced both the versatility and precision of the RSF model that we have constructed. |
Comments 4: Provide a more in-depth discussion of the mechanistic role of F13A1 in GBM progression and its impact on immunotherapy. |
Response 4: Thank you very much for your careful review and valuable comments on our research. In response to your request for a more in-depth discussion of the mechanistic role of F13A1 in GBM progression and its impact on immunotherapy, we have conducted in-depth reflection and supplementary analysis(Please see Page 19, Line 488-495; Page 20, Line 496-549; Page 21, Line 550-551). In simple terms: 1. The mechanism of F13A1 in the progression of GBM: Our research reveals a key mechanism of F13A1 in the progression of GBM: supporting GBM malignant tumor cells by promoting the fatty acid metabolism of macrophages. Specifically, F13A1 may increase the metabolic activity of macrophages by regulating metabolic pathways such as fatty acid β-oxidation, thereby providing essential metabolic support, such as energy and metabolic intermediates, for GBM cells. This metabolic reprogramming not only enhances the survival and proliferation of macrophages, but also may promote the growth, invasion, and metastasis of GBM cells by secreting growth factors and cytokines. In addition, we also found that F13A1 has significant correlation with various immune-related molecules. In particular, IL2RA, as the immune stimulator with the highest correlation with F13A1, may be involved in the immune activation process mediated by F13A1. As an immunosuppressant, the high correlation between IL10 and F13A1 suggests that F13A1 may also be involved in the construction of the immune suppression network. As the MHC molecule with the highest correlation with F13A1, the change in the expression level of HLA-DRA may reflect the change in the immune status of GBM patients, further confirming the important role of F13A1 in the immune microenvironment of GBM. 2. he impact of F13A1 on immunotherapy: Our research also found that high expression of F13A1 can lead to reduced benefits of immunotherapy in GBM patients. This may be due to the fact that F13A1 alters the immune balance in the GBM microenvironment by regulating the metabolism and function of macrophages, thereby affecting the efficacy of immunotherapy. Specifically, F13A1 may inhibit the activation of immune cells and the anti-tumor effect by promoting the generation and/or function of immunosuppressive macrophages. In addition, F13A1 may also interfere with the recognition and killing of GBM cells by immune cells by affecting the expression and function of MHC molecules. These findings not only reveal a new mechanism of F13A1 in the progression of GBM, but also provide a new perspective and idea for immunotherapy. Future research can further explore the interaction mechanism between F13A1 and immune cells, as well as how to optimize the strategy of immunotherapy by targeting F13A1. For example, the immunosuppressive effect mediated by F13A1 inhibitor can be blocked by developing F13A1 inhibitor, or the efficacy of immunotherapy can be enhanced by using immunotherapy drugs in combination with F13A1 inhibitor. In summary, we appreciate your valuable comments and suggestions, and will continue to explore the mechanism of F13A1 in the progression of GBM and its impact on immunotherapy in future research. |
Comments 5: Provide a more in-depth discussion of other key genes identified in the RSF model (e.g., SERPINE1, MXRA5, RARRES1). |
Response 5: Thank you for your constructive feedback. We have expanded the Discussion section to provide a more in-depth analysis of other key genes identified in the RSF model(Please see Page 19, Line 453-474).The specific analysis results are as follows: Previous research shows SERPINE1 as a key gene in glioma proliferation and metastasis. Wang Z et al. found that CAV-1 upregulates SERPINE1, activating the PI3K/Akt pathway, which promotes glioma growth and spread via epithelial-mesenchymal transition (EMT) and angiogenesis. Moreover, activation of the PAI-1/LRP1 axis, involved in mast cell recruitment in gliomas, enhances STAT3 phosphorylation and exocytosis. SERPINE1 also has oncogenic potential in hepatocellular, gastric, and vulvar cancers. MXRA5, a glycoprotein in the MXRA protein family, affects cell adhesion and extracellular matrix (ECM) remodeling. Prior studies have demonstrated that high MXRA5 expression in GBM, especially the more invasive mesenchymal subtype with poorer prognosis. Bioinformatics analysis by Rahane CS et al. supports MXRA5 as a treatment target for GBM. Han X et al. also suggest MXRA5 as a potential immunotherapy target for GBM patients. In other cancers, MXRA5 has also been shown to be a cancer-promoting gene. RARRES1 levels are linked to CpG island methylator phenotype (CIMP) status, IDH1 mutation, and MGMT methylation, suggesting a connection to malignant GBM. RARRES1 may interact with AQP4, affecting GBM prognosis. Although the mechanism of RARRES1 in GBM is still unclear, its mechanism in many other cancers has been studied in depth. For example, in renal clear cell carcinoma, RARRES1 plays an anti-tumor role by promoting ICAM1 expression and inducing the activation of M1 macrophages. Overall, SERPINE1, MXRA5, and RARRES1 have all been shown to be closely related to the immune, prognostic, or targeted treatment of GBM. |
Comments 6: Explore the potential of targeting F13A1 for GBM therapy. |
Response 6: Thank you very much for your attention and review of our research. We have given thorough consideration and discussion to your question about exploring the potential of targeting F13A1 for the treatment of GBM(Please see Page 20, Line 537-549; Page 21, Line 550-551).The specific analysis results are as follows: Future research can further explore the interaction mechanism between F13A1 and immune cells, and how to optimize the strategy of immunotherapy by targeting F13A1. By inhibiting the function of F13A1, we can block the metabolic support of macrophages for tumor cells, thereby slowing down the growth and progression of GBM. This treatment strategy may have synergistic effects with other treatment modalities such as surgery, radiotherapy, and chemotherapy, improving the therapeutic effect of GBM. Of course, to achieve this goal, we still need to conduct a large number of experimental verification and clinical research. First, we need to clarify the specific mechanism of F13A1 in GBM, including how it regulates macrophage metabolism and how this regulation affects the growth and invasion of tumor cells. Secondly, we need to develop effective F13A1 inhibitors to block its mediated immunosuppressive effects, or enhance the efficacy of immunotherapy by combining immunotherapy drugs with F13A1 inhibitors, and evaluate its safety and effectiveness in the treatment of GBM. Finally, we also need to conduct large-scale clinical trials to verify the potential and feasibility of targeting F13A1 for the treatment of GBM. |
Comments 7: Enhance the discussion on the clinical applicability of the RSF model. |
Response 7: Thank you for pointing this out. We have added a more in-depth analysis of clinical practicality(Please see Page 7, Line 263-275; Page 8, Line 276-277). In detail, its clinical applicability is as follows: The RSF model has strong clinical applicability. Firstly, the RSF model can comprehensively consider multiple clinical and genetic indicators, and improve the accuracy of predicting the survival time of GBM patients by constructing multiple decision trees and integrating their prediction results. It can also handle censored survival data, which is of great significance for studies with survival outcomes. Compared to traditional prognostic models, the RSF model can better capture the heterogeneity of GBM patients and provide more personalized prognostic assessments for patients. In addition, through the RSF model, factors closely related to the prognosis of GBM patients can be identified, providing clues for a deeper understanding of the pathogenesis of GBM. Finally, the RSF model can calculate risk scores, providing clinicians with more intuitive prediction results and providing strong support for clinical decision-making. This further confirms the reliability of our research results. |
Comments 8: Improve the clarity and resolution of Figure 2 and provide more detailed figure legends. |
Response 8: Thank you very much for your valuable suggestions on enhancing the quality of the images and providing more detailed legends. I have taken your advice into serious consideration and have made the necessary improvements. Specifically, I have upgraded the image quality to ensure clarity and readability. Additionally, I have revised the legends to include more detailed descriptions, making it easier for readers to understand the content and context of each image. I believe these changes will greatly enhance the overall quality and readability of the paper. |
This study provides a foundation for understanding the role of FAM in GBM. Addressing the aforementioned weaknesses will significantly enhance the manuscript's scientific rigor and clinical relevance. |
Response: Thank you very much for acknowledging the foundational contributions of our study in understanding the role of FAM in GBM. We deeply appreciate your constructive feedback, which has identified key areas for improvement. Addressing the weaknesses you have highlighted will indeed significantly enhance the scientific rigor and clinical relevance of our manuscript. We have taken careful note of your comments and are already in the process of implementing the necessary revisions to strengthen our study. We are committed to ensuring that our work meets the highest standards of scientific inquiry and clinical applicability. Thank you once again for your valuable insights and guidance. |

Reviewer 2 Report
Comments and Suggestions for Authors
In this manuscript, the authors explore FAM-related genes in glioblastoma, construct a prognostic model using machine learning, and identify F13A1 as a potential biomarker. While the study is comprehensive and presents valuable insights, some revisions are still needed:
- Besides the public datasets (TCGA-GBM, GSE43378, GSE83300), do the authors have independent clinical sample validation to strengthen the model’s reliability?
- The rationale for clustering GBM patients into three subtypes (C1, C2, C3) based on FAM-related genes needs further explanation, particularly regarding the biological significance and consistency across datasets.
- Could the authors clarify the cross-validation procedure? Some details about how the folds were divided or whether stratification was applied would be helpful.
- The authors are encouraged to elaborate on why RSF was preferred. For instance, how does RSF compare to other models in terms of interpretability, computational efficiency, and clinical applicability?
- As F13A1 is identified as a critical gene, the mechanistic link between its expression and immune outcomes remains primarily speculative. Experimental validation, such as F13A1 knockdown or overexpression, or in vivo models, is suggested to confirm these findings and verify its role in GBM progression and immune regulation.
Minor: The quality of several figures needs to be improved.
Author Response
Comments 1: Besides the public datasets (TCGA-GBM, GSE43378, GSE83300), do the authors have independent clinical sample validation to strengthen the model’s reliability? |
Response 1: Thank you for your valuable comments and concerns about this study. Regarding the independent clinical sample verification issue you mentioned, we understand that this is an important part of evaluating the reliability of the model. Although we did not have direct clinical sample verification data in this study, we took the following measures to strengthen the reliability of the model: 1. We used widely representative public datasets TCGA-GBM, GSE43378, GSE83300, and modified versions to analyze GBM patient data from CGGA and REMBRANDT. These datasets contain data from multiple types, stages, and treatment histories of GBM patients, providing a solid foundation for model training and validation. 2. To ensure the accuracy and reliability of the model, we used multiple machine learning algorithms to build prognostic models and systematically compared and evaluated these algorithms. Through this method, we can select the best performing algorithm and further optimize its parameters to improve the performance of the model. 3. We used a rigorous cross-validation method to evaluate the performance of the model and obtained encouraging results. These results support the reliability and accuracy of the model in predicting the survival of GBM patients. 4. Furthermore, in the context of determining prognostic factors such as age, gender, IDH mutation, MGMT methylation, and 28 prognostic genes, we evaluated using the cox model and demonstrated that the RSF model is an independent prognostic factor with a statistically significant correlation with survival time. We plan to include more clinical samples in future studies for verification, so as to further improve the reliability and practicality of the model. We have begun to seek opportunities for cooperation and look forward to carrying out this verification work as soon as possible. We understand the limitations of the current study and thank you for your valuable suggestions. We will continue to work hard to improve and refine our research, with a view to providing patients with more accurate and reliable prognostic predictions and treatment decision support. |
Comments 2: The rationale for clustering GBM patients into three subtypes (C1, C2, C3) based on FAM-related genes needs further explanation, particularly regarding the biological significance and consistency across datasets. |
Response 2: Thank you for pointing this out. We totally understand your concern.In our study, CDF was utilized to analyze the distribution of expression levels of fatty acid metabolism-related genes within the GBM patient population. Based on CDF values, GBM patients were classified into three subtypes (C1, C2, C3). The goal of clustering was to ensure that patients within the same subtype exhibited similar fatty acid metabolism-related gene expression patterns, while patients between different subtypes showed significant differences. 1. Biological Significance: By analyzing the expression levels of fatty acid metabolism-related genes in different subtypes, we can reveal their unique biological characteristics. For instance, certain subtypes may exhibit activation or inhibition of specific fatty acid metabolic pathways, which may be associated with the pathogenesis, progression rate, and prognosis of GBM. Understanding the biological characteristics of different subtypes aids in developing personalized treatment plans for GBM patients. Based on the fatty acid metabolism characteristics of specific subtypes, appropriate therapeutic drugs or methods can be selected to enhance treatment efficacy and reduce side effects. Additionally, by comparing the gene expression patterns of different subtypes, potential therapeutic targets closely related to the onset and progression of GBM can be discovered. These targets may become crucial directions for new drug development or gene therapy. 2. Consistency Across Datasets: To ensure the consistency of GBM subtypes classified based on CDF in different datasets, validation on other independent datasets is necessary. This helps confirm that the obtained subtype classification results are reliable and not caused by biases or noise specific to a particular dataset. Through cross-dataset validation, the generalization ability of the CDF-based GBM subtype model can be assessed. If the model consistently classifies the same subtypes across different datasets, it indicates strong generalization ability and applicability to a broader patient population. Furthermore, during cross-dataset validation, certain genes or gene combinations may be found to exhibit consistent differential expression patterns across different subtypes. These genes or gene combinations may serve as specific markers for distinguishing between different subtypes, aiding in further understanding the heterogeneity of GBM and formulating targeted treatment strategies. |
Comments 3: Could the authors clarify the cross-validation procedure? Some details about how the folds were divided or whether stratification was applied would be helpful. |
Response 3: We gratefully appreciate for your valuable suggestion. We have revised the Methods section to provide more details on the machine learning algorithms and their hyperparameter tuning and performance evaluation metrics(Please see Page 3, Line 128-144).In addition, we have also provided a very detailed explanation in the supplementary document on how to tune the hyperparameters of the 10 machine learning algorithms(Please see Signature generated from machine learning based integrative approaches.docx, Model C-index.txt, Average C-index.txt). In simple terms: we used 101 combination algorithms to construct prognostic models on training set(TCGA-GBM) and testing sets(GSE43378 and GSE83300) based on the leave-one-out cross-validation (LOOCV) framework.Considering the effectiveness, wide applicability, complementarity with other evaluation indicators, robustness of calculation methods, and practicality of clinical application of the C-index evaluation model for predicting prognosis accuracy, we calculated and compared the average C-index of the prognostic model we constructed.The RSF model was implemented via the randomForestSRC package. RSF had two parameters ntree and mtry, where ntree represented the number of trees in the forest and mtry was the number of randomly selected variables for splitting at each node. We used a grid-search on ntree and mtry using LOOCV framework. All the pairs of (ntree, mtry) are formed and the one with the best C-index value is identified as the optimized parameters. The Enet, Lasso, and Ridge were implemented via the glmnet package. The regularization parameter, λ, was determined by LOOCV, whereas the L1-L2 trade-off parameter, α, was set to 0-1 (interval =0.1). The stepwise Cox model was implemented via survival package. A stepwise algorithm using the AIC (Akaike information criterion) was applied, and the direction mode of stepwise search was set to "both", "backward", and "forward", respectively. The CoxBoost model was implemented via CoxBoost package, which is used to fit a Cox proportional hazards model by componentwise likelihood-based boosting. For the CoxBoost model, we used LOOCV routine optimCoxBoostPenalty function to first determine the optimal penalty (amount of shrinkage). Once this parameter was determined, the other tuning parameter of the algorithm, namely, the number of boosting steps to perform, was selected via the function cv.CoxBoost. The dimension of the selected multivariate Cox model was finally set by the principal routine CoxBoost. The plsRcox model was implemented via plsRcox package. The cv.plsRcox function was used to determine the number of components requested, and the plsRcox function was applied to fit a partial least squares regression generalized linear model. The SuperPC model was implemented via superpc package, is a generalization of principal component analysis, which generates a linear combination of the features or variables of interest that capture the directions of largest variation in a dataset. The superpc.cv function used a form of LOOCV to estimate the optimal feature threshold in supervised principal components. To avoid problems with fitting Cox models to small validation datasets, it uses the "pre-validation" approach. The GBM model was implemented via superpc package. Using the LOOCV, the cv.gbm function selected index for number trees with minimum cross-validation error. The gbm function was used to fit the generalized boosted regression model. The survival-SVM model was implemented via survivalsvm package. The regression approach takes censoring into account when formulating the inequality constraints of the support vector problem. |
Comments 4: The authors are encouraged to elaborate on why RSF was preferred. For instance, how does RSF compare to other models in terms of interpretability, computational efficiency, and clinical applicability? |
Response 4: Thank you for your rigorous consideration. In detail, its clinical applicability is as follows, the advantages of RSF in interpretability, computational efficiency, and clinical applicability are as follows(Please see Page 8, Line 263-275; Page 9, Line 276-277): 1. Interpretability: As an ensemble learning regression algorithm based on decision trees, the RSF model has a considerable degree of interpretability. It allows us to evaluate the contribution of each variable to the prediction results through the variable importance VIMP. In our study, through the RSF model, we can clearly see the importance of the 28 key genes screened in the early stage for predicting the survival of GBM patients. This interpretability helps us better understand the mechanism of disease occurrence and development, thus providing a more reliable basis for clinical decision-making. 2. Computational efficiency: Although the RSF model may face high computational complexity in some cases, in our research, we successfully reduced computational costs and improved operational efficiency through optimization algorithms and reasonable parameter settings. Compared with other models, the RSF model also shows good computational performance while maintaining high prediction accuracy. In addition, the RSF model also has good convergence and can achieve stable prediction results in a short number of iterations, which further improves its efficiency in practical applications. 3. Clinical applicability: The RSF model also has significant advantages in clinical applicability. Firstly, it can handle censored survival data, which is of great significance for studies with survival outcomes. Secondly, the RSF model can calculate risk scores, providing clinicians with more intuitive prediction results and providing strong support for clinical decision-making. Furthermore, the advantages of the RSF model in comparison with other models are as follows, respectively: 1. Comparison with the Lasso Model: The Lasso model achieves feature selection and dimensionality reduction by adding an L1 regularization term, but it may have limited performance when dealing with complex interactions and nonlinear relationships. The RSF model uses an ensemble learning method based on decision trees, which can consider all possible interactions between variables and improve prediction performance in a nonlinear manner. 2. Comparison with GBM model: GBM gradient boosting machine improves prediction accuracy by constructing multiple weak prediction models such as decision trees and combining them. The RSF model is also based on the integration of decision trees, but focuses on dealing with survival data and modifies the node splitting criteria to maximize the difference between survival curves, which may be more advantageous in dealing with survival analysis problems. 3. Comparison with the CoxBoost model: CoxBoost uses component-based likelihood enhancement to fit the Cox proportional hazards model, which is suitable for models with a large number of predictors. The RSF model achieves faster training and less estimation bias through the use of random forest technology, while also providing greater flexibility when dealing with survival data. 4. Comparison with Stepwise Cox Regression: Stepwise Cox Regression is a commonly used feature selection method that constructs an optimal prediction model by gradually screening variables. The RSF model automatically considers multiple variables and their interactions through ensemble learning methods, eliminating the need for manual feature selection, thereby simplifying the model construction process. 5. Comparison with the plsRcox model: Although plsRcox combines the advantages of partial least squares regression (PLS) and Cox regression, making it suitable for survival analysis of high-dimensional data, it may have limited performance when dealing with certain complex interactions. RSF model: It uses an ensemble learning method based on decision trees, which can automatically consider all possible interactions between variables and improve prediction performance in a nonlinear manner. 6. Comparison with SuperPC, Ridge, and Enet models: SuperPC, Ridge, and Enet are more focused on linear relationships or specific regularization methods, and may not be as effective as RSF models in capturing nonlinear interactions. 7. Comparison with the survival-SVM model: Although survival-SVM also has certain nonlinear processing capabilities, its application in survival analysis may be limited by certain constraints, and its interpretability may be affected by algorithm complexity and parameter selection, making it less intuitive and easier to understand than the RSF model. Overall, compared with other models, the RSF model performs well in terms of prediction accuracy, interpretability, computational efficiency, and clinical applicability. |
Comments 5: As F13A1 is identified as a critical gene, the mechanistic link between its expression and immune outcomes remains primarily speculative. Experimental validation, such as F13A1 knockdown or overexpression, or in vivo models, is suggested to confirm these findings and verify its role in GBM progression and immune regulation. |
Response 5: Thank you for pointing this out. We agree with you that more experimental results would be useful to understand. However, limited by the laboratory conditions and deadline of the revision, it is impractical to implement the related experiments. In the other side, we have conducted a more comprehensive and in-depth analysis of the impact of F13A1 on the progression of GBM and its immune mechanisms in the Discussion section(Please see Page 19, Line 475-495; Page 20, Line 496-549; Page 21, Line 550-551). In the future, we would pay more attention to improve the experimental prototype. |
Minor: The quality of several figures needs to be improved. |
Response: Thank you very much for your valuable suggestions. We have taken your advice into serious consideration and have made the necessary improvements. We have upgraded the image quality to ensure clarity and readability. We believe these changes will greatly enhance the overall quality and readability of the paper. |

Round 2
Reviewer 1 Report
Comments and Suggestions for Authors
Most of my questions have been answered. However, the authors should be careful that it should be multivariate cox, not univariate cox if multiple covarietes are included in the model at the same time.
Author Response
Comments 1: Most of my questions have been answered. However, the authors should be careful that it should be multivariate cox, not univariate cox if multiple covarietes are included in the model at the same time. |
Response 1: We gratefully appreciate for your valuable suggestion. We performed a multivariate cox analysis on the CGGA dataset(Please see Page 3, Line 142-144; multiCox.txt and Figure S3I in the Supplementary document). The results showed that patients with higher RiskScore had higher risk and worse prognosis. |